# SHAPE-guided RNA structure homology search and motif discovery

Edoardo Morandi[1], Martijn J. van Hemert [2] & Danny Incarnato [1✉]

The rapidly growing popularity of RNA structure probing methods is leading to increasingly large amounts of available RNA structure information. This demands the development of efficient tools for the identification of RNAs sharing regions of structural similarity by direct comparison of their reactivity profiles, hence enabling the discovery of conserved structural features. We here introduce SHAPEwarp, a largely sequence-agnostic SHAPE-guided algorithm for the identification of structurally-similar regions in RNA molecules. Analysis of Dengue, Zika and coronavirus genomes recapitulates known regulatory RNA structures and identifies novel highly-conserved structural elements. This work represents a preliminary step towards the model-free search and identification of shared and conserved RNA structural features within transcriptomes.

[1] Department of Molecular Genetics, Groningen Biomolecular Sciences and Biotechnology Institute (GBB), University of Groningen, Groningen, The Netherlands. [2] Department of Medical Microbiology, Molecular Virology Laboratory, Leiden University Medical Center, Leiden, The Netherlands. ✉email: d.incarnato@rug.nl

A key need in RNA biology is the ability to identify structurally-homologous (or structurally-similar) RNAs. One of the most widely used methods for RNA structure homology search is Infernal[1]. Infernal relies on probabilistic models, called covariance models[2] (CMs), which describe both the consensus sequence and secondary structure of a family of related RNAs, and it uses a dynamic programming algorithm to identify putative structurally-homologous RNAs that can be optimally aligned to the CMs. While this method is extremely accurate and powerful, it depends on the availability of both a high-quality alignment of related RNA sequences and an accurate secondary structure model for the RNA family of interest. Although determining the structure of an RNA is a nontrivial task, the introduction over the last decade of a wide range of methods, mostly based on the coupling of RNA structure probing and high-throughput sequencing, has significantly improved our ability to interrogate the structure of thousands of RNA molecules in a single experiment[3]. Among these, methods based on chemical probing are probably the most popular. Chemical probing allows to rapidly query RNA structures in a variety of contexts, including the complex cellular environment, by taking advantage of specific chemical probes that can selectively react with RNA nucleotides, in a way that is directly correlated with their structural context[4]. Dimethyl sulfate[5] (DMS), a nucleobase-specific probe for unpaired A and C bases, and Selective 2′-hydroxyl acylation analyzed by primer extension (SHAPE) reagents[6–8], which react with the 2′-OH of the ribose moiety on structurally-flexible residues, are widely employed probes as they can readily permeate cell membranes, allowing to study RNA structures in their physiological context. The readout of these experiments is a reactivity profile that informs on the reactivity of each nucleotide in the RNA to the employed chemical probe. Such reactivity profiles can be used in conjunction with free energy minimization algorithms to derive an experimentally-informed structure model of the RNA of interest. This is typically done by converting the experimentally-determined reactivities into pseudo-free-energy

contributions, that are used to adjust the thermodynamics parameters (known as nearest-neighbor model[9]) in order to constrain the RNA structure prediction and yield a structure model that better agrees with the observed reactivity profile[10], although many alternative approaches have been also proposed[11–14]. However, the accuracy of the inferred structure models largely relies on a multitude of factors, including the employed chemical probe, the approach used to incorporate the experimentally-determined reactivities into the structure prediction, and a number of method-specific analysis parameters. As a consequence, deriving high-quality RNA secondary structure models is a nontrivial task, as minimal changes in the analysis protocol might significantly affect the reliability of the inferred RNA structure model. Furthermore, the growing interest in RNA as a target for small molecule drugs underscores the importance of this challenge, as well as the need for tools able to evaluate if a candidate RNA structure can be selectively targeted, or, in other words, whether a candidate target is sufficiently unique within the transcriptome[15]. To fill this gap, we here introduce SHAPEwarp, a model-free and sequence-agnostic method for the identification of structurally-similar RNA regions in a database of chemical probing-derived reactivity profiles. By applying SHAPEwarp to a number of SHAPE datasets for viral RNA genomes we demonstrate that, not only SHAPEwarp is able to recover known functional RNA structures, but most importantly that it can be used to drive the identification of previously unidentified conserved RNA structure elements.

## Results and discussion

The approach used by SHAPEwarp (see Supplementary Note 1) is inspired by the BLAST algorithm[16] and it builds on top of two widely used methods for similarity search in time series data: Mueen's Algorithm for Similarity Search (MASS)[17] and dynamic time warping (DTW). In a first step (Fig. 1a), all the possible query kmers of a user-defined length are enumerated, filtering out those corresponding to low structure complexity regions. Regions

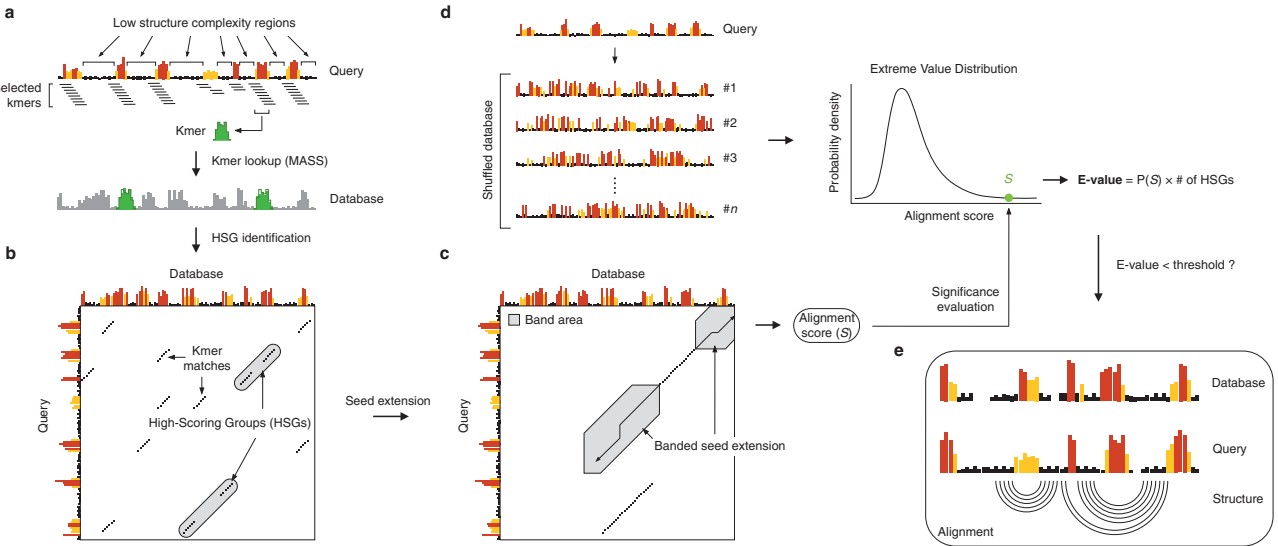

**Fig. 1 Schematic of the SHAPEwarp algorithm. a** All the possible kmers along a query SHAPE reactivity profile are enumerated, discarding those having low structure complexity. Retained kmers are looked up into the database via the MASS algorithm. **b** A matrix is built storing the coordinates of each query kmer and its matches in the database, allowing the grouping of kmers lying on the same diagonal into high scoring groups (HSGs). **c** Each HSG is used as the seed to begin the bidirectional extension of an alignment. Banding of the alignment restricts the search space to a maximum number of bases around the diagonal. The alignment stops when the score (S) drops below a certain threshold for more than a certain number of bases. **d** In parallel, the same query is searched against a database of shuffled SHAPE reactivity profiles. The scores of these alignments are used to build the null distribution, further allowing to estimate the probability of obtaining an alignment score ≥S. From this the E-value of the alignment can be estimated. **e** Optionally, significant alignments can be further analyzed for the presence of a conserved structure by exploiting the RNAalifold algorithm.

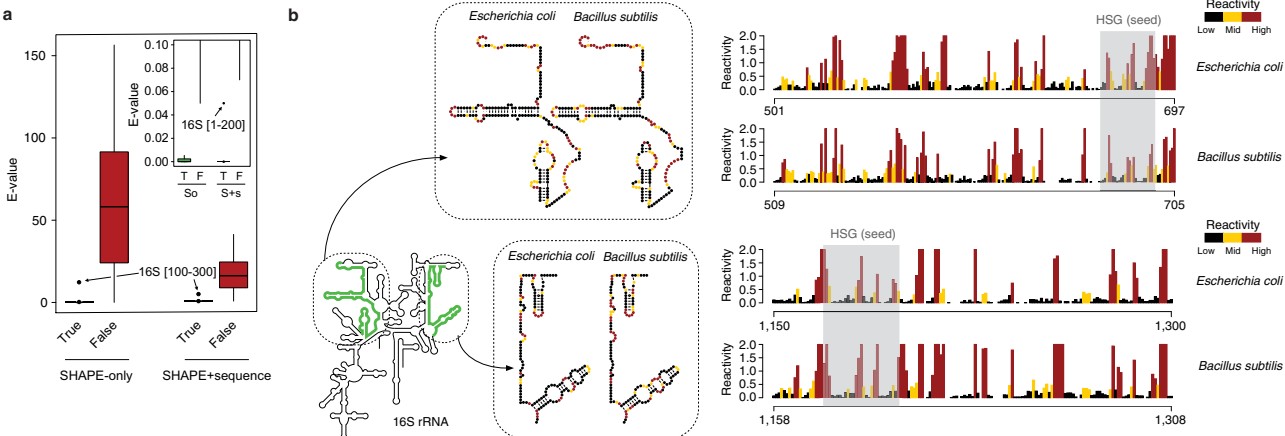

**Fig. 2 Validation of SHAPEwarp. a** Box-plot depicting the distribution of E-values for true (T) and false (F) matches for *E. coli* 16S rRNA searched against *B. subtilis* 16S/23S rRNAs, both in SHAPE-only (So) and SHAPE + sequence (S + s) mode. Boxes span the 25th to the 75th percentile. The center represents the median. Values below the 25th percentile − 1.5 times the IQR, or above the 75th percentile + 1.5 times the IQR, represent outliers and are reported as dots. The inset shows a zoom-in view of the box-plot for E-values between 0 and 0.1. Sample sizes are as follows: n = 15 (SHAPE only) and n = 16 (SHAPE + sequence) for true matches, and n = 1338 (SHAPE only) and n = 457 (SHAPE + sequence) for false matches. **b** Sample alignments for two matching regions between the 16S rRNAs of *E. coli* and *B. subtilis*, as identified by SHAPEwarp. SHAPE reactivities have been capped to 2. The high scoring group (HSG), constituting the seed of the alignment, is shaded in gray. The insets show the same two regions in their structural context.

of low structure complexity are defined as those showing little or no disparity in the distribution of SHAPE reactivities, as it would be expected for kmers entirely falling inside a base-paired or unpaired region of the RNA. Filtering of these kmers is performed by imposing a minimum cutoff on their Gini coefficient[18,19]. Kmers passing the complexity cutoff are then looked up in a database of reactivity profiles via MASS, a highly optimized algorithm for time series subsequences all-pairs-similarity search (TSAPSS) in just $O(n^2)$. Additionally, the percent GC content of the kmer and its matches are evaluated and matches having GC content deviating by more than a certain percentage from that of the query kmer, are discarded. Indeed, analysis of multiple sequence alignments for known RNA families from RFAM suggests that the local percent GC content of structurally-related RNAs tends to show only limited variation (median GC% difference: 15.10%, 13.02%, 11.51%, and 9.98% respectively for kmer lengths of 8, 10, 12 and 15 nucleotides) (Supplementary Fig. 1). Kmer matches lying on the same diagonal, within a maximum distance from each other, are then grouped into high scoring groups (HSGs) (Fig. 1b). HSGs are essentially sub-segments of a query-database pair, that share a high degree of similarity and can be aligned without gaps. Ungapped HSGs are extended upstream and downstream by using a banded semi-global alignment algorithm, that incorporates both features of the Gotoh's Smith–Waterman algorithm with affine gap penalties[20] and of DTW. The extension process is stopped when the score drops below a certain threshold for more than a certain number of nucleotides (Fig. 1c). A null model is then built by searching the same query in a database of randomly shuffled reactivity profiles (Fig. 1d) and used to estimate the E-value of each database match[21]. For those matches passing the user-defined E-value threshold, the aligned reactivity profiles can then be used to model the underlying structure by using the RNAalifold algorithm[22], a widely used method for inferring consensus secondary structures from RNA alignments (Fig. 1e). Kmer matching and alignment extension parameters (see Methods) were optimized by using in vivo SHAPE-MaP data for the 23S rRNA obtained by probing *E. coli* and *B. subtilis* with 2-aminopyridine-3-carboxylic acid imidazolide[7] (2A3), a powerful SHAPE reagent we recently developed[7], and validated on the 16S rRNA. *E. coli* 16S rRNA was first split into 200 nucleotides-long

windows, with 100 nucleotides overlap, and used to query *B. subtilis* rRNAs (Fig. 2a). With the selected parameter set and by employing an E-value threshold of 0.01 for the SHAPE-only search, or 0.005 when also accounting for sequence (SHAPE + sequence), SHAPEwarp had a false discovery rate (FDR) of 0 and sensitivity of ~87% (13/15 windows recovered). The distribution of E-values for true matches (median SHAPE only: 3.4e−5; median SHAPE + sequence: 0) showed a clear separation (p value SHAPE only: 6.1e−11; p value SHAPE + sequence: 2.2e−11; two-sided Wilcoxon Rank Sum test) with respect to the distribution of E-values for false matches (median SHAPE only: 58.15; median SHAPE + sequence: 15.47) (Fig. 2b). We performed an additional validation by using a previously published SHAPE-MaP dataset for four Dengue virus (DENV) serotypes[23], probed in virio with NAI. Viral genomes present the advantage of having extremely conserved architectures, hence allowing to estimate the FDR of our method. We performed any of the six possible comparisons of the four DENV serotypes (both SHAPE only and SHAPE + sequence), each time by using one serotype as the query dataset and the other serotype as the database. Once again, SHAPEwarp showed a remarkably low FDR (SHAPE only: ~1.3%; SHAPE + sequence: ~0.5%). In all comparisons but one (DENV-2 vs. DENV-4, SHAPE only), SHAPEwarp matched, to different extents, the 5′ and 3′ UTRs of the genomes, known to carry several key RNA regulatory elements (Supplementary Fig. 2a). The resulting alignments (along with the corresponding SHAPE profiles) were further fed into RNAalifold. We then extracted the individual structure elements and used them as input for *cm-builder*, a method we recently developed to allow the automated identification of conserved RNA structures supported by significant covariation[24,25]. Our pipeline combining SHAPEwarp and cm-builder recovered the most common Flavivirus regulatory elements[26] (SLA, cHP, xrRNA, DB, and CRE) (Extended Data Fig. 1b). It further recovered another highly-conserved element within the capsid coding region (DCS-PK), previously reported to regulate genome cyclization[27]. CMs derived from this pipeline performed comparably well to RFAM manually-curated CMs[28]. Notably, the CM for the xrRNA element outperformed the one from RFAM (RF03547), as it respectively matched ~75% and ~70.5% of the scanned ZIKV and DENV genomes, versus 0% and ~1.5% respectively matched by the RFAM model (Supplementary

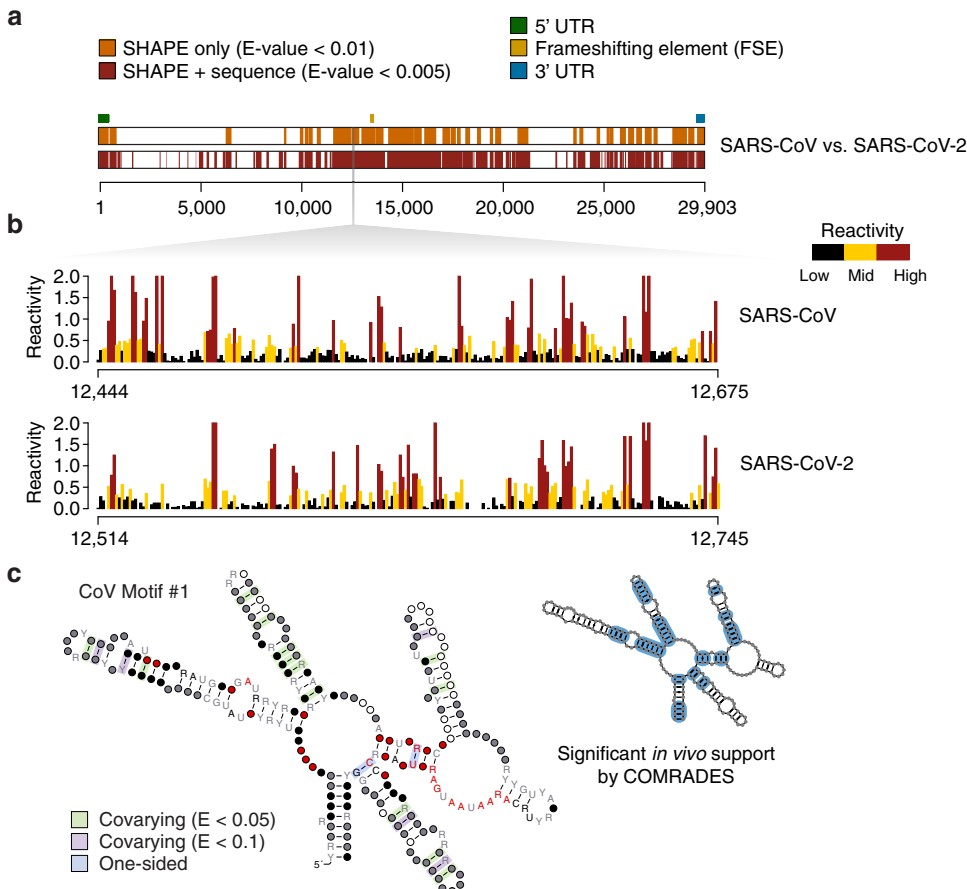

**Fig. 3 SHAPEwarp identifies novel highly-conserved viral RNA structures. a** Significant matches between SARS-CoV (query) and SARS-CoV-2 (database), identified by SHAPEwarp, either in SHAPE only (orange) or SHAPE + sequence (red) mode. The relative position of known RNA structure elements (5′ UTR, FSE and 3′ UTR) is indicated. **b** Aligned SHAPE reactivity profiles for one of the identified structurally-conserved regions (CoV Motif #1). SHAPE reactivities have been capped to 2. **c** Structure model for CoV Motif #1. Structure was generated using R2R[43]. One-sided covariations were inferred from R2R output. Base pairs showing significant covariation (as determined by R-scape) are boxed in green (*E* value < 0.05) and violet (*E* value < 0.1) respectively. The inset illustrates base pairs having significant RNA–RNA chimera support from in vivo COMRADES, boxed in blue.

Fig. 2b). Encouraged by the performances of SHAPEwarp, we next sought to apply it to drive the discovery of novel RNA structure elements in viral genomes. To this end we used a recently published icSHAPE dataset[29] (RT stop-based readout) comparing the Asian and African lineages of Zika virus (ZIKV) by in vivo probing with NAI[6] and we further generated two new datasets by probing the structure of both SARS and SARS-CoV-2 coronaviruses (CoVs) with 2A3. SHAPEwarp identified extensive structural homology between the compared genomes for both ZIKV (SHAPE only: ~61.8%; SHAPE + seq: ~80.9%) and CoVs (SHAPE only: ~41.9%; SHAPE + seq: ~61.2%). The identified structurally-homologous regions encompassed known structural elements (i.e., 5′ and 3′ UTRs for both ZIKV and CoVs, as well as the frame-shifting element for CoVs). Besides known structures in the UTRs, application of our combined SHAPEwarp+cm-builder pipeline resulted in the identification of eight novel conserved RNA elements, of which five in ZIKV and three in CoVs (Fig. 3 and Supplementary Figs. 3–12). It is worth noticing that we only selected structures having extensive covariation support, hence this analysis likely represents an underestimate of the actual number of functionally-conserved RNA structures in these genomes. Importantly, besides showing significant covariation support (as determined by R-scape[30]), all eight structures but one (ZIKV Motif #4) showed significant in vivo support by direct RNA–RNA interaction capture via COMRADES[31,32] in infected host cells, further underscoring their functional

significance. Lack of significant support by COMRADES for ZIKV Motif #4, is most likely the consequence of its limited size. In a recent study we introduced DRACO[25], a method for the deconvolution RNA structure ensembles from DMS-MaPseq[33] data, and used it to identify multiple regions within the SARS-CoV-2 genome folding into two alternative structures. Comparison of SARS-CoV-2 structurally-heterogeneous regions with the three structurally-conserved structure elements here identified showed that CoV Motif #3 partially overlaps with one of the regions identified by DRACO to form two alternative conformations. The deconvolved reactivity profile for the major conformation (59.0 ± 1.5%) well agrees with the structure here identified by our SHAPEwarp + cm-builder pipeline. Importantly, analysis of the minor conformation (41.0 ± 1.5%) with cm-builder revealed that weaker but significant covariation and COMRADES support exist also for this structure (Supplementary Fig. 6), hence underscoring the potential relevance of this structural switch. Altogether, these results demonstrate the ability of SHAPEwarp to drive the identification of structurally-homologous RNA regions, in a model-free fashion. When considering all the SHAPE datasets here analyzed, SHAPEwarp exhibited an overall FDR of ~1.6% and ~0.3% respectively for SHAPE only and SHAPE + sequence searches. SHAPEwarp robustness is largely independent from which SHAPE reagent or readout strategy is employed. For instance, the extensive similarity detected for the ZIKV genomes underscores this point.

Indeed, while optimal SHAPEwarp alignment parameters were derived from MaP data, the icSHAPE data for ZIKV genomes was generated using an RT-stop-based readout strategy. Furthermore, we evaluated SHAPEwarp's performances with both NAI and 2A3-derived profiles, generated by either RT-stop or mutational profiling. Search for NAI-derived profiles in a database of 2A3-derived profiles further supports this notion, showing a sensitivity of ~95.5% and an FDR of ~2% (Supplementary Fig. 13).

We further evaluated the performances of SHAPEwarp with queries of different lengths (100, 150, and 250 nt) and compared the results to those previously obtained with 200 nucleotide-long queries. To this end, we calculated the fraction of correctly matched 200 nucleotide-long windows (passing the minimum E-value cutoff), that overlapped with at least one correctly matched window at each of the evaluated query lengths, imposing a minimum overlap of 25% (calculated with respect to the smaller window). For SHAPE only, this resulted into overlaps of 54%, 86% and 97.2% respectively for 100, 150, and 250 nucleotide-long queries. For SHAPE + sequence, the overlap was way more extensive, with 92.1%, 99.3%, and 99.5% respectively for 100, 150, and 250 nucleotide-long queries. The marked difference in overlap with shorter queries observed between SHAPE only and SHAPE + sequence searches can be easily explained by the very low probability of finding by chance a match having both similar SHAPE reactivity and nucleotide sequence. Thus, the difference between the score of a true match and the distribution of alignment scores from the shuffled database tends to be way more significant when sequence identity is rewarded as well. This difference can be mitigated by increasing the E-value cutoff for shorter queries. Indeed, by simply increasing the E-value cutoff from 0.01 to 0.05, the overlap for 100 nt-long queries increases from 54% to 76%, while keeping the FDR below 5%.

In summary, we have here introduced SHAPEwarp, a SHAPE-driven sequence-agnostic method for the identification of structurally-similar RNA regions. We can anticipate that SHAPEwarp will greatly facilitate the discovery of shared and conserved RNA structural features within transcriptomes.

## Methods

**SHAPEwarp algorithm**. The SHAPEwarp algorithm is implemented in Perl and Rust. SHAPEwarp comes with three components: SHAPEwarp, swKmerLookup and swBuildDb. The SHAPEwarp module allows searching one or more queries in a database of SHAPE reactivities (or base-pairing probabilities). The swKmerLookup module (invoked by SHAPEwarp) is responsible of performing both the kmer lookup and kmer grouping into HSGs. Finally, the swBuildDb module allows generating SHAPEwarp-compliant databases of SHAPE reactivities (or base-pairing probabilities). A complete description of the algorithm is provided in Supplementary Note 1.

**Data retrieval**. SHAPE-MaP data for *E. coli* and *B. subtilis* rRNAs, DENV and SARS-CoV-2 (probed with NAI) were obtained from the Gene Expression Omnibus (GEO) database (GSE154563, GSE106483 and GSE151327). Prenormalized icSHAPE data for ZIKV was obtained from Li et al.[29]. COMRADES datasets for ZIKV and SARS-CoV-2 virus in living infected host cells were respectively obtained from ArrayExpress (E-MTAB-6427) and GEO (GSE154662).

**Analysis of SHAPE-MaP data**. All the analysis steps, from reads alignment to data normalization, were performed using RNA Framework[34]. All tools referenced in the following paragraphs are distributed as part of the RNA Framework suite (https://github.com/dincarnato/RNAFramework). The following parameters were used: rf-map (parameters: -b2 -cq5 20 -ctn -cmn 0 -mp '--very-sensitive-local', using Cutadapt v2.1[35], Bowtie v2.3.5.1[36] and SAMtools v1.9[37]), rf-count (parameters: -m -rd), and rf-norm (parameters: -sm 3 -nm 3 -mm 1 -n 1000). For the analysis of DENV SHAPE-MaP data, the -n parameter of rf-norm was lowered to 500, to ensure the inclusion of genome boundaries. Only replicate 1 of each experiment was used, given the exceptionally high inter-sample correlation.

**Selection and optimization of folding parameters**. For base-pair probability calculation and structure modeling, the following previously determined slope and intercept pairs were used: 1.0 and −0.4 for 2A3[7] and 1.1 and 0.0 for NAI[23]. For

NAI-N3, a reference structure including the known 5′ and 3′ UTR elements of ZIKV was built for the Asian strain (PRVABC59) and optimal values 2.4 and −0.6 were identified by grid search using the rf-jackknife tool (parameters: -rp "-md 600 -nlp" -x) and ViennaRNA v2.4.14[38], and by selecting the prediction achieving the highest sensitivity.

**Cell culture and SARS-CoV/SARS-CoV-2 infection**. Infection with SARS-CoV and SARS-CoV-2 was conducted as previously described[24,39]. Briefly, Vero E6 cells were cultured in DMEM (Lonza, cat. 12-604F), supplemented with 8% FCS (Bodinco), 2 mM L-glutamine, 100 U/mL of penicillin and 100 μg/mL of streptomycin (Sigma Aldrich, cat. P4333-20ML). Cells were infected with either SARS-CoV or SARS-CoV-2 in EMEM (Lonza, cat. 12-611F) supplemented with 25 mM HEPES, 2% FCS, 2 mM L-glutamine, and penicillin/streptomycin. 16 h after the infection, infected cells were harvested by trypsinization, followed by resuspension in EMEM supplemented with 2% FCS, washed with 50 mL 1X PBS, and then resuspended in 1 mL of QIAzol Lysis Reagent (Qiagen, cat. 79306). All experiments with infectious SARS-CoV/SARS-CoV-2 were performed in a biosafety level 3 facility at the LUMC.

**Total RNA extraction and in vitro folding**. To 1 mL of infected cells in QIAzol Lysis Reagent, 200 μl of chloroform were added. The sample was vigorously vortexed for 15 sec and then incubated for 2 min at room temperature, after which it was centrifuged for 15 min at 12,500 × g (4 °C). The upper aqueous phase was collected in a clean 2 mL tube, supplemented with 1 mL (~2 volumes) of 100% ethanol, and then loaded on a Monarch® RNA Cleanup Kit column (NEB, cat. T2030L). In vitro folding was carried out as previously described[24,40]. Briefly, ~5 μg of total RNA from infected cells were first depleted of ribosomal RNAs using the RiboMinus™ Eukaryote System v2 (ThermoFisher Scientific, cat. A15026), as per manufacturer instructions. rRNA-depled RNA in a volume of 39 μl was denatured at 95 °C for 2 min, then transferred to ice for 1 min. 10 μl of ice-cold 5X RNA Folding Buffer [500 mM HEPES pH 7.9; 500 mM NaCl] supplemented with 20 U of SUPERase•In™ RNase Inhibitor (ThermoFisher Scientific, cat. AM2696) were added. RNA was then incubated for 15 min at 37 °C to allow secondary structure formation. Subsequently, 1 μl of 500 mM MgCl$_2$ (pre-warmed at 37 °C) was added and RNA was further incubated for 15 min at 37 °C to allow tertiary structure formation.

**Probing of SARS-CoV and SARS-CoV-2 RNA**. For probing of RNA, 2A3 was added to a final concentration of 100 mM (assuming a stock concentration of 1 M). An equal volume of DMSO was added to the control samples. Samples were then incubated at 37 °C for 5 min. Reactions were quenched by the addition of 1 volume DTT 1 M and then purified on Monarch® RNA Cleanup Kit columns.

**SHAPE-MaP analysis of SARS-CoV and SARS-CoV-2 RNA**. SHAPE-MaP library preparation was conducted as previously described[24], with minor changes. Probed RNA was first fragmented to a median size of 150 nt by incubation at 94 °C for 8 min in RNA Fragmentation Buffer [65 mM Tris-HCl pH 8.0; 95 mM KCl; 4 mM MgCl$_2$], then purified on Monarch® RNA Cleanup Kit columns and eluted in 8 μl NF H$_2$O. Eluted RNA was end repaired by treatment with 1 U of rSAP (NEB, cat. M0371L) at 37 °C for 30 min, plus 5 min at 70 °C to heat-inactivate the enzyme, followed by treatment with 20 U of T4 PNK (NEB, cat. M0201L), in the presence of 1 mM ATP, at 37 °C for 1 h. 50 ng of end-repaired RNA were first ligated to 10 pmol of a pre-adenylated 3′ adapter (rApp-AGATCGGAAGAGCACACGTCT-SpC3) using 200 U of T4 RNA Ligase 2 truncated KQ (NEB, cat. M0373L), in the presence of 12.5% PEG-8000, for 1 h at 25 °C. 10 pmol of a 5′ RNA adapter (CUACACGACGCUCUUCCGAUCU) were then ligated using 30 U of T4 RNA Ligase 1 (NEB, cat. M0437M), in the presence of 1 mM ATP and 8% PEG-8000, at 25 °C for 1 h. The adapter-ligated RNA was then supplemented with 10 pmol of RT primer (AGACGTGTGCTCTTCCGATCT), incubated at 70 °C for 5 min and immediately transferred to ice for 1 min. Reverse transcription reactions were conducted in a final volume of 10 μl. Reactions were supplemented with 2 μl 5X RT Buffer [250 mM Tris-HCl pH 8.3; 375 mM KCl], 1 μl DTT 0.1 M, 0.5 μl dNTPs 10 mM, 0.5 μl MnCl$_2$ 120 mM, 20 U SUPERase•In™ RNase Inhibitor and 100 U SuperScript II (ThermoFisher Scientific, cat. 18064014). Reactions were incubated at 42 °C for 2 h. RNA was degraded by addition of 1 μl NaOH 5 N, followed by incubation at 95 °C for 3 min. After purification, barcodes were introduced by PCR, using NEBNext® High-Fidelity 2X PCR Master Mix (NEB, cat. M0541L) and NEBNext® Multiplex Oligos for Illumina® Dual-Index primers (NEB, cat. E7780S).

**Optimization of SHAPEwarp parameters**. To optimize SHAPEwarp parameters, we took advantage of in vivo 2A3 probing data for *E. coli* and *B. subtilis* 23S rRNAs we previously generated[7] and used a structurally-informed alignment of the 23S rRNAs from the two species (generated using Infernal[1]) as a reference. For the calibration of kmer lookup and kmer grouping parameters, a 200 nt-long window was slid along the *E. coli* 23S rRNA, with an offset of 100 nt, and used to query the *B. subtilis* 23S rRNA, using any possible combination of the following parmeters: maxKmerDist (10, 20, 30); kmerLen (8, 10, 12, 15); minKmers (1, 2, 3); kmerMinComplexity (0.2, 0.3, 0.4); maxReactivity (1, 1.5 for SHAPE reactivities; 1 for base-pairing probabilities); kmerMaxMatchEveryNt (100, 200, 500). A query was

considered to represent a true positive match if at least one HSG overlapped by at least 1 nt the expected position in the reference alignment. Any other match was regarded as false positive. The set of parameters having the highest ratio of true positive matches to the average number of false positives per window, was picked. For the calibration of alignment parameters, we proceeded in two steps. First, by using a subset of 15 true positive HSGs to seed the alignment of $15 \times 200$ nt-long queries, alignment extension was performed using any possible combination of the following parameters: alignMatchScore (min: −2, −1.5, −1, −0.5, 0; max: 1, 1.5, 2, 2.5, 3, 3.5, 4, 4.5, 5, 6); alignMismatchScore (min: −8, −7.5, −7, −6.5, −6, −5.5, −5, −4.5, −4; max: −2, −1.5, −1, −0.5, 0); maxReactivity (1, 1.5, 2 for SHAPE reactivities; 1 for base-pairing probabilities). During the first step, the following parameters were set to a fixed value: alignGapOpenPenal (−12); alignGapExtPenal (−5); alignMaxDropOffRate (0.7); alignMaxDropOffBases (5). After having computed the score of the true alignments, the reactivity values around each HSG in the database were randomly shuffled 100 times and aligned to the original query. The null distribution built by using the scores of the alignments to the shuffled database was then used to calculate the $p$ values of the true alignments, by using the extreme value distribution (see Supplementary Note 1). The top ten sets, resulting into the lowest average $p$ value for the searched queries, were retained. In step 2, HSG extension was repeated, by using the possible values for alignMatchScore, alignMismatchScore and maxReactivity from the top ten selected sets, and any possible combination of the following parameters: alignGapOpenPenal (−8, −12, −14, −16); alignGapExtPenal (−5, −7, −9); alignMaxDropOffRate (0.5, 0.6, 0.7, 0.8); alignMaxDropOffBases (5, 6, 7, 8, 9, 10). Once again, the $p$ value of each alignment was calculated using the aforementioned procedure, and the set of parameters resulting in the lowest average $p$ value for the searched queries was selected. This procedure resulted in the following parameter sets: SHAPE reactivities [kmerLen: 15; minKmers: 2; maxKmerDist: 30; kmerMinComplexity: 0.3; kmerMaxMatchEveryNt: 200; alignMatchScore: −0.5, 2; alignMismatchScore: −6, −0.5; alignGapOpenPenal: −14; alignGapExtPenal: −5; alignMaxDropOffRate: 0.8; alignMaxDropOffBases: 8; maxReactivity: 1] and base-pairing probabilities [kmerLen: 15; minKmers: 3; maxKmerDist: 10; kmerMinComplexity: 0.4; kmerMaxMatchEveryNt: 200; alignMatchScore: 0, 3; alignMismatchScore: −7, 0; alignGapOpenPenal: −12; alignGapExtPenal: −9; alignMaxDropOffRate: 0.7; alignMaxDropOffBases: 6; maxReactivity: 1].

**Building the SHAPEwarp databases**. Databases were generated using the swBuildDb module (parameters: --chunkSize 1000), starting from RNA Framework's normalized SHAPE reactivity XML files. For bacterial rRNAs, a single database was built including both the 16S and 23S rRNAs. For DENV, ZIKV and SARS-CoV-2 genomes, a separate database was built for each strain. Database shuffling was performed in 10 nt blocks. This approach allows generating more realistic profiles, as it is likely that nearby bases will reside within a similar structural context (i.e., a loop), hence preserving the structural features of the profile. 100 shuffles were performed for each database entry and a chunk of a maximum size of 1000 nt was extracted from each shuffled entry and used to build the shuffled database. This random sampling step does not introduce any bias (provided that the searched query is shorter than the chosen chunk size) while reducing the computation time.

**Searching for structurally-homologous regions with SHAPEwarp**. Query transcripts were first split into 200 nt-long windows, with 100 nt overlap. Each query window was searched against the database using the previously optimized search parameters. When accounting for sequence in the alignment (SHAPE + sequence), the following additional parameters were set: --alignScoreSeq --alignSeqMatchScore 0.5 --alignSeqMismatchScore −2. Consecutive matching windows were then merged and alignments were evaluated for the presence of conserved structure elements by analyzing the SHAPEwarp-generated alignment with RNAalifold v2.4.14. SHAPEwarp-generated alignments were then randomly shuffled 100 times, and the resulting alignments were analyzed with RNAalifold. Shuffling was performed in 3 column blocks, followed by random shuffling of the columns within each block. A null distribution was then built using the free energy of the structures inferred from the shuffled alignments. The free energy of the true alignment was then converted into a Z-score and the corresponding probability was determined by using the normal distribution. The above-described analysis steps are integrated into SHAPEwarp (parameters: --evalAlignFold --inBlockShuffle). Structures having a $p$ value < 0.05 and a base-pair support (measured as the fraction of canonical base pairs supported by each sequence in the alignment) of at least 0.75, were retained for covariation analysis. For FDR estimation, matches falling at the same relative genomic position (with a tolerance of ±2%) in the query and database genomes were considered to represent true matches, while any other match was assumed to represent a false match. When searching NAI-derived profiles in a database of 2A3-derived profiles, the exact genomic position was required to match (0% tolerance).

**Identification of RNA structure elements supported by significant covariation**. To evaluate the conservation of the predicted structure elements, we further improved cm-builder (https://github.com/dincarnato/labtools), an automated pipeline we have previously introduced[24,25], built on top of Infernal v1.1.3. Briefly,

a first covariance model (CM) was built from the structure inferred by RNAalifold, using the cmbuild module. For structures <100 nt, a 100 nt window centered on the structure was extracted and used to build the CM. After calibrating the CMs using the cmcalibrate module, it was used to search for RNA homologs in a database composed of either all the non-redundant *Orthocoronavirinae* or *Flaviviridae* complete genome sequences from the ViPR database[41], using the cmsearch module. Only matches from the sense strand were kept and a relaxed E-value threshold of 10 was used at this stage to select potential homologs. Matches were required to fall at the same relative position within their respective genomes, with a tolerance of 3.5% for coronavirus genomes (roughly corresponding to a maximum allowed shift of 1050 nt in a 30 kb genome) and 2% for flavivirus genomes (roughly corresponding to a maximum allowed shift of 220 nt in a 11 kb genome). Furthermore, matches retaining less than 55% of the canonical base pairs from the original structure and truncated hits covering <50% of the structure were discarded. The whole procedure was repeated a maximum of 3 times. As compared to the original cm-builder approach, we further introduced an additional evaluation step. Briefly, at each iteration of the algorithm, the resulting Stockholm alignment was analyzed with R-scape v1.4.0[30] and APC-corrected G-test statistics to evaluate the presence of significantly covarying base pairs, using a relaxed E-value of 0.1. If the detected number of significant covariations was lower than that detected at the previous iteration, the alignment from the previous iteration was reported. Furthermore, the search was aborted if no significant covariation was detected after 2 iterations. Retained alignments were then manually inspected and only complex structures (i.e., multiway junctions) having covariation support for multiple helices were retained. The selected best alignment (the one with the most significant covarying base pairs) was then refactored by removing non-structured regions at either sides and used to build the final CM.

**Search for conserved structures in related genomes**. The CMs built by the cm-builder pipeline were used to search for matches in the same non-redundant set of viral genomes used to build the CMs, with the cmscan module. To further evaluate the base-pair support of the identified matches, the alignment between the CM and the match on the target genome returned by cmscan was analyzed to calculate the fraction of canonical base pairs supporting the structure.

**In vivo structure support by COMRADES**. Each COMRADES dataset consisted of two to three biological replicates, each one composed of a control (C) and the actual COMRADES sample (S). A reference was built on all human transcripts from refGene, plus the sequence of either the ZIKV or SARS-CoV-2 genomes, using STAR v2.7.1a[42] (parameters: --runMode genomeGenerate --genomeSAindexNbases 12), and reads were aligned to the reference using the same (parameters: --runMode alignReads --outFilterMultimapNmax 100 --outSAMattributes All --alignIntronMin 1 --scoreGapNoncan −4 --scoreGapATAC −4 --chimSegmentMin 15 --chimJunctionOverhangMin 15). Prior to parsing, replicates were merged. Chimera support for the base pairs inferred from RNAalifold analysis of SHAPEwarp alignments was calculated using the rf-duplex tool (parameters: -st -mh 5 -xr 1e9 -eo). Briefly, alignments were filtered to discard reads having more than one gap, as well as ungapped reads and reads aligning to the human transcriptome, and the total number of reads per experiment was calculated ($C_{tot}$ and $S_{tot}$). To assess whether a candidate base-pair $i–j$ was significantly enriched in the COMRADES sample with respect to the control sample, we calculated the number of chimeras for which one side of the chimera encompassed base $i$ and the other side encompassed base $j$, for both samples ($C_{i–j}$ and $S_{i–j}$). Significance of the enrichment was then assessed using a one-tailed binomial test, with parameters $k = S_{i–j}$, $n = S_{tot}$, and $p = C_{i–j}/C_{tot}$. Base pairs having $p$ value < 0.05 were considered to have in vivo support.

**Reporting summary**. Further information on research design is available in the Nature Research Reporting Summary linked to this article.

## Data availability

Sequencing data used in this study have been deposited to the Gene Expression Omnibus (GEO) database, under the accession number GSE189259. Additional processed files (normalized SHAPE reactivity profiles, SHAPEwarp databases and queries needed to reproduce the analyses detailed in the manuscript, Stockholm alignments and CMs for the structure elements identified in this work) are available at http://www.incarnatolab.com/datasets/SHAPEwarp_Morandi_2022.php.

## Code availability

The source code of SHAPEwarp has been deposited in GitHub, https://github.com/dincarnato/SHAPEwarp (https://doi.org/10.5281/zenodo.6327165). A user documentation, along with explanation of the different parameters, is available on Read the Docs, https://shapewarp-docs.readthedocs.io/.

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

## Acknowledgements

We would like to thank T. Marinus (University of Groningen) for initial critical discussion of the SHAPEwarp algorithm. D.I. was supported by the Dutch Research Council (NWO) as part of the research program NWO Open Competitie ENW—XS with project number OCENW.XS3.044 and by the Groningen Biomolecular Sciences and Biotechnology Institute (GBB, University of Groningen). M.J.H. was supported by the Leiden University Fund (LUF), the Bontius Foundation and donations from the crowdfunding initiative "wake up to corona".

## Author contributions

Project conceptualization: E.M. and D.I.; SARS-CoV and SARS-CoV-2 manipulations: M.J.H.; SHAPEwarp algorithm design and implementation: E.M. and D.I.; Bioinformatics, structure modeling and data analysis: E.M. and D.I.; Writing: D.I.

## Competing interests

The authors declare no competing interests.
