## [Peer Review File · Nature Communications]

REVIEWER COMMENTS

Reviewer #1 (Remarks to the Author):

In the manuscript 'SHAPE-guided RNA structure homology search and motif discovery', the authors propose a novel method for the determination of structurally homologous RNAs that employs structure probing information. The SHAPEwarp approach comprises aspects of sequence similarity search methods, comparison of reactivity profiles from RNA structure probing databases, and consensus structure folding. The authors apply their method to viral data sets, showcasing SHAPEwarp's potential to characterize known regulatory elements in Flaviviruses and Coronaviruses.

The paper is well written and the overall message is clear. The text details algorithmic and methodologic aspects of SHAPEwarp's dual approach of incorporating BLAST-like and reactivity profile database lookup, paired with consensus structure prediction. This is an interesting and potentially powerful approach that will facilitate screening for structures RNAs in high-throughput probing data. As such, the proposed method represents a substantial improvement for the field of RNA structural biology.

There are only a few issues that should be addressed before publication:

- line 45: Some readers might not be familiar with 'low structural complexity'. The authors might want to explain this briefly.

- line 46: Please explain the Gini coefficient and provide a reference.

- line 94: There might be more regulatory elements besides those mentioned. Therefore, I would change the wording to 'the most common', instead of 'all the known'.

- line 99: What is meant by 'outperforming' one covariance model by another one? Please explain.

- line 414: The authors might discuss the impact of alternative window and step sizes. Is there significant overlap of target elements when running SHAPEwarp with window sizes of say 100nt/150nt/200nt/250nt? Larger window sizes might not be relevant, as most known functional RNAs are in the range between 50/60nt and 250nt

As a general remark, I would appreciate if the authors made all candidate Stockholm files of the SHAPEwarp Zika/Dengue/Coronavirus screen publicly available on GitHub.

Reviewer #2 (Remarks to the Author):

In this Communication, Incarnato and co-workers report a novel algorithm called "SHAPEwarp", and

describe its implementation in a pipeline with their cm builder to identify novel RNA structural homologs (motifs/domains) primarily using experimental data of in vivo and in virio chemical reactivities. They call this method “model-free” to contrast it with my model-based approaches for finding motifs, such as RNAbob. My opinion is that model-free approaches are sorely needed in the field; it’s not enough any more to just find more examples of known motifs. Their algorithm involves identifying kmers along identical diagonal elements, and is inspired by BLAST. They demonstrate success by identifying known motifs in vRNA and in rRNA. They then apply the method to a set of corona and Zika viruses and identify both known and potentially novel motifs. Nicely, these motifs are backed up by more experimental data, namely crosslinks from COMRADES. This promising model-free RNA motif finder will likely be important for the field. Following are some comments for the authors to consider.

Major Comments

1.) It is stated that the pipeline/algorithm performs equally well with RT-stops and mutational profiling (MaP).

(a) Which data in the manuscript come from MaP? Which from RT-stops?

(b) MaP is notable for providing multiple reactivities on a single read, which can allow one to look for multiple, mutually exclusive structure for a given motif. Have the authors tried Rouskin’s DREEM protocol to see if multiple structures are predicted in the identified regions? One possibility is that their discovered motifs have a single structure, which is why they are conserved, and that would be helpful to know. Another is that the structure switches between two folds and that too is conserved.

2.) On a related note, riboSNitches are RNAs that change structure in the presence of SNPs. Are there SNPs for the Zika and CoV in the structurally homologous regions and do they interfere with folding? If so, then the biological importance of the SNPs is potentially high. There are a number of programs available to carry out riboSNitch predictions.

3.) The authors should indicate how they will make their work available to the non-bioinformatician.

Other Comments:

1.) Lines 59-61. The extension process is of interest. Presumably the length of the extension varies from motif to motif. What are the extension values found for length: average length, standard deviation of length, and range of lengths. Are the 5’- and 3’-end lengths extended independently? Does it tend to be longer on the 5’ or 3’ ends? Is the extension tested 1 nucleotide at a time or are there bigger jumps?

2.) Line 74. “When also accounting for sequence.” What does this mean exactly? The limited GC variation from line 53?

3.) Line 80. Writing is ambiguous. The values of 58.15 and 15.47 look to be for FALSE in Fig 2a but come across as for TRUE in the writing.

4.) Where is the ZIKV motif #2 in the figures? Seems like it should be between SI fig #4 and #5.

5.) End on line 124 might be clearer if worded, “from which SHAPE reagent is employed”

Reviewer #3 (Remarks to the Author):

This work by Morandi et al., developed a very important tool in identifying the structure homology using SHAPE probing data. Previously, the structure conservation was only measured by sequence alignments. This SHAPEwarp filled the gap between sequence alignment and experimental probing data. This method will advance the whole research field, allowing the identification of conserved functional structure motif. The methodology development and validations are very clear. The manuscript is well written.

I only have one comment:

The SHAPE profiles for individual RNAs are highly dependent on the sequencing depth. Is there any requirement on the SHAPE reactivity profiles (eg., reads/nt?) for running the SHAPEwarp alignment?

Reviewer #1 (Remarks to the Author):

In the manuscript 'SHAPE-guided RNA structure homology search and motif discovery', the authors propose a novel method for the determination of structurally homologous RNAs that employs structure probing information. The SHAPEwarp approach comprises aspects of sequence similarity search methods, comparison of reactivity profiles from RNA structure probing databases, and consensus structure folding. The authors apply their method to viral data sets, showcasing SHAPEwarp's potential to characterize known regulatory elements in Flaviviruses and Coronaviruses.

The paper is well written and the overall message is clear. The text details algorithmic and methodologic aspects of SHAPEwarp's dual approach of incorporating BLAST-like and reactivity profile database lookup, paired with consensus structure prediction. This is an interesting and potentially powerful approach that will facilitate screening for structures RNAs in high-throughput probing data. As such, the proposed method represents a substantial improvement for the field of RNA structural biology.

We would like to thank the reviewer for their positive comments and appreciation of our work.

There are only a few issues that should be addressed before publication:

- line 45: Some readers might not be familiar with 'low structural complexity'. The authors might want to explain this briefly.
- line 46: Please explain the Gini coefficient and provide a reference.
- line 94: There might be more regulatory elements besides those mentioned. Therefore, I would change the wording to 'the most common', instead of 'all the known'.
- line 99: What is meant by 'outperforming' one covariance model by another one? Please explain.

We thank the reviewer for the suggestions. We have now addressed all the points in the revised manuscript. Concerning the last point, particularly, by "outperforming" we mean that the covariation model built by our pipeline respectively matched ~75% and ~70.5% of the scanned ZIKV and DENV genomes, versus 0% and ~1.5% respectively matched by the RFAM model. We have now clarified this in the revised manuscript.

- line 414: The authors might discuss the impact of alternative window and step sizes. Is there significant overlap of target elements when running SHAPEwarp with window sizes of say 100nt/150nt/200nt/250nt? Larger window sizes might not be relevant, as most known functional RNAs are in the range between 50/60nt and 250nt

We thank the reviewer for raising this important point. To address it, we have now performed the search with different window sizes (100/150/250 nt, respectively slid in 50/75/125 nt increments). We then calculated the fraction 200 nt windows resulting into significant matches (from our original analysis), that overlapped by at least 25% with a significant match at query lengths of 100 nt, 150 nt and 250 nt.

For the SHAPE-only search, with default parameters, at an E-value cutoff of 0.01, the overlap with the analysis performed in 200 nt windows was: 54% at 100 nt, 86% at 150 nt and 97.2% at 250 nt. However, this analysis does not account for the fact that search parameters might need to be adjusted for shorter queries. For instance, with shorter queries, the E-value threshold needs to be increased to reflect the fact that the expectation of finding by chance an alignment with a score $\geq x$ is higher than it would be for a longer query. Indeed, by simply increasing the E-value cutoff to 0.05, the overlap at 100 nt jumps from 54% to 76%, while the FDR remains <5%.

For the SHAPE+sequence search, instead, with default parameters, at an E-value cutoff of 0.005, the overlap is way more extensive: 92.1% at 100 nt, 99.3% at 150 nt and 99.5% at 250 nt. This is somewhat expected, as the probability of finding by chance a match with both a similar SHAPE profile and a similar sequence is very low (hence, the difference between the score of a true match and the distribution of scores from shuffled alignments is way more significant). We have now commented on this point in the revised manuscript.

As a general remark, I would appreciate if the authors made all candidate Stockholm files of the SHAPEwarp Zika/Dengue/Coronavirus screen publicly available on GitHub.

We apologize if this detail was missing in the original manuscript. The Stockholm files are already publicly available on our website, at the address: https://www.incarnatolab.com/datasets/SHAPEwarp_Morandi_2022.php. We have amended the "Data availability" statement to clarify this.

Reviewer #2 (Remarks to the Author):

In this Communication, Incarnato and co-workers report a novel algorithm called "SHAPEwarp", and describe its

implementation in a pipeline with their cm builder to identify novel RNA structural homologs (motifs/domains) primarily using experimental data of in vivo and in virio chemical reactivities. They call this method “model-free” to contrast it with my model-based approaches for finding motifs, such as RNAbob. My opinion is that model-free approaches are sorely needed in the field; it’s not enough anymore to just find more examples of known motifs. Their algorithm involves identifying kmers along identical diagonal elements, and is inspired by BLAST. They demonstrate success by identifying known motifs in vRNA and in rRNA. They then apply the method to a set of corona and Zika viruses and identify both known and potentially novel motifs. Nicely, these motifs are backed up by more experimental data, namely crosslinks from COMRADES. This promising model-free RNA motif finder will likely be important for the field. Following are some comments for the authors to consider.

We would like to thank the reviewer for their appreciation of our work and for the positive remarks on the importance of having a model-free approach for RNA structure homology search and discovery.

Major Comments

1.) It is stated that the pipeline/algorithm performs equally well with RT-stops and mutational profiling (MaP).

(a) Which data in the manuscript come from MaP? Which from RT-stops?

All the data generated by our lab comes from SHAPE-MaP (so, the data used for parameter calibration on *E. coli* and *B. subtilis* rRNAs [PMID: 33398343] and the probing data for SARS-CoV and SARS-CoV-2 [this study]), while the data for ZIKV, generated in a previous study (PMID: 30472207), comes from icSHAPE, that is RT-stop-based.

(b) MaP is notable for providing multiple reactivities on a single read, which can allow one to look for multiple, mutually exclusive structure for a given motif. Have the authors tried Rouskin’s DREEM protocol to see if multiple structures are predicted in the identified regions? One possibility is that their discovered motifs have a single structure, which is why they are conserved, and that would be helpful to know. Another is that the structure switches between two folds and that too is conserved.

We thank the reviewer for highlighting this point. Indeed, the structure signal observed in structure probing experiments might arise from a heterogeneous mixture of structures. Unfortunately, there is a number of issues connected with deconvoluting the underlying structure ensemble by using SHAPE data and DREEM (or similar approaches):

a) DREEM can only analyze a region of an RNA when the reads fully cover that region, therefore it is not suited for the analysis of typical MaP experiments in which the RNA is randomly fragmented and sequenced with short reads (the approach we used here for SHAPE-MaP library preparation). This is one of the reasons that led our lab to develop DRACO (PMID: 33619392). DRACO has been thoroughly validated and shown to outperform DREEM in terms of sensitivity and accuracy.

b) One of the main requirements of DREEM, DRACO and related approaches is a high sequencing depth (5000-10000X or greater), way above the minimum coverage required for accurate SHAPE-MaP analysis (1000X). Therefore, at the current coverage of the SHAPE-MaP dataset here reported, it would be impossible to perform the deconvolution analysis.

c) The previous point is further exacerbated when considering that SHAPE-MaP experiments have in general higher background as compared to DMS-MaP experiments. Indeed, DREEM, DRACO and related approaches have been developed for, and solely validated on, DMS-MaP data. Therefore, to date, it is unknown whether they would perform adequately on SHAPE-MaP data as well.

However, to address the important point raised by the reviewer, we took advantage of MaP data we previously generated by DMS probing of the SARS-CoV-2 genome and analyzed with DRACO (PMID: 33619392). Our previous analysis (please refer to Supplementary Fig. 17 of PMID: 33619392) showed that both regions corresponding to CoV Motifs #1 and #2 (pos. 12,514-12,745 and pos. 23,645-23,928) are not among the regions of the SARS-CoV-2 identified by DRACO to form 2 alternative conformations. For what concerns CoV Motif #3 (pos. 26,495-26,861) instead, it partially overlaps with one of the regions found by DRACO to form two alternative mutually-exclusive conformations (pos. 26,570-26,751). Notably, the reactivities for the major conformation (accounting for $\sim 59 \pm 1.5\%$) of the population, well agree with the structure identified by our combined SHAPEwarp+cm-builder analysis (and depicted in Supplementary Fig. 4 of this manuscript). The minor conformation, instead, adopts a different arrangement of the left-most multiway junction. This alternative conformation is also supported, to a lower extent, by significant covariation and by COMRADES. This conformation has now been included in Supplementary Fig. 5 of the revised manuscript.

2.) On a related note, riboSNitches are RNAs that change structure in the presence of SNPs. Are there SNPs for the Zika and CoV in the structurally homologous regions and do they interfere with folding? If so, then the biological importance of the SNPs is potentially high. There are a number of programs available to carry out riboSNitch

predictions.

We agree with the referee that evaluating the presence of potential riboSNitches would be potentially important to reveal the importance of SNPs overlapping with the identified structurally-homologous regions. To evaluate this possibility, we proceeded as it follows. For SARS-CoV-2, we obtained available sequences from GISAID (<https://www.gisaid.org/>) and we extracted a random sample of 100,000 sequences. Similarly, for ZIKV, we obtained all the available sequences from the VIPR database (<https://www.viprbrc.org/>), having as host human (822 sequences). We then used Infernal's cmscan and the covariance models we previously built to locate the position of the structure elements in each of the genomes. The corresponding sequence was then extracted (after discarding sequences having large insertions/deletions or more than 10% Ns) and, for each structure element, we built a multiple sequence alignment. We then identified at each position of the alignment variants occurring in >1% of the sequences. Notably, of the 53 variants identified, 49 involved C->U and G->A transitions. As these variants are not expected to negatively affect base-pairing, we disregarded them. Also, it is worth pointing out that, in many cases, more than one variant co-occurred within the same sequence for a given structure element. Therefore, evaluating the impact of each variant independently would be incorrect.

We focused instead on the remaining 4 variants corresponding to two C->G transversions in CoV Motif #3, one A->C transversion in CoV Motif #2 and one A->C transversion in ZIKV Motif #2. To evaluate the potential impact of these variants, we used two approaches. We first calculated the partition function for the sequences and determined the base-pairing probabilities within the Boltzmann ensemble. Then, we calculated the Z centroid structure (the structure having all the base-pairs with >50% probability in the ensemble), previously shown to be among the top-performing metrics for evaluating the structure disrupting effect of riboSNitches (PMID: 22759654), and compared the Z centroid structures for each two variants of the structure elements by calculating the geometric mean of their PPV and sensitivity. The comparisons returned values in the range 0.95-1. The same analysis, conducted directly on the MFE structure, gave the same result. As an alternative approach, we directly calculated the correlation between the base-pairing probabilities across the ensembles of each two variants and, once again, the returned correlations were in the range of 0.95-1. Altogether these analyses suggest that the analyzed variants have little or no disruptive effect on the ensembles of these structure elements. Therefore, we decided not to include the results of this analysis in the revised manuscript.

3.) The authors should indicate how they will make their work available to the non-bioinformatician.

We agree with the reviewer that this represents an important point. The standalone program will be provided via GitHub, and accompanied by an extensive and detailed manual, available at <https://shapewarp-docs.readthedocs.io/>. We have now updated the "Code availability" statement of the revised manuscript accordingly. It is also worth pointing out that the output generated by SHAPEwarp is very user-friendly. Particularly, sequence alignments, as well as plots of aligned reactivity profiles (like the ones shown in this manuscript) are automatically generated.

In the near future, we also plan to make SHAPEwarp available as a web server for the non-bioinformaticians. This involves having a dedicated machine with sufficient computational resources, that is being currently set up. However, we believe this is outside of the scope of the present manuscript, and it will be published elsewhere.

Other Comments:

1.) Lines 59-61. The extension process is of interest. Presumably the length of the extension varies from motif to motif. What are the extension values found for length: average length, standard deviation of length, and range of lengths. Are the 5'- and 3'-end lengths extended independently? Does it tend to be longer on the 5' or 3' ends? Is the extension tested 1 nucleotide at a time or are there bigger jumps?

The extension procedure is performed in 2 steps (analogously to the BLAST algorithm). In the first step, the seed is extended "upstream" (in the 5' direction). The extension uses a semi-global alignment approach. This means that, the upper-left corner of the scoring matrix is initialized to the score of the seed match, rather than 0. We apologize with the reviewer if this step was not clearly described in the Supplementary Note 1. We have now amended the text and formula #12 defining the matrix initialization. The scoring matrix is then populated (using the standard dynamic programming approach) as long as the score does not drop below a minimum threshold. At that point, the 5' seed extension step is terminated. In the second step, the seed is extended "downstream" (in the 3' direction). The approach is the same as detailed for the 5' extension, with the sole difference that, this time, the upper-left corner of the scoring matrix is initialized to score of the seed match, plus the score of the 5' extension (if positive). Therefore, the two extensions are independently performed.

In the above plots, we have considered the seed extensions for all the significant alignments performed in this study. When looking at the distributions of extension lengths in either the 5' or 3' direction (panel A), it is possible to notice that the extensions span approximately the same range, although a small difference exists in the median extension length, with the 3' extension being slightly skewed towards longer extensions (median: 19 for 5' vs. 24 for 3'). Nonetheless, this can be easily explained when considering that the queries here searched were 200 nt-long and that the positioning of the “best” seeds (the ones resulting into productive extensions) was slightly skewed towards the 5' end (panel B, median: 92). This is most likely the consequence of the relatively small number of seed extensions here considered (~1000). For larger numbers, the median of the distribution is expected to tend to 100 (the exact center of a 200 nt-long query).

2.) Line 74. “When also accounting for sequence.” What does this mean exactly? The limited GC variation from line 53?

This refers to the fact that SHAPEwarp can perform the alignment by either accounting solely for the SHAPE reactivities (“SHAPE-only”), or by accounting also for the underlying nucleotide sequence (“SHAPE+sequence”). This is done by adding an extra reward (or penalty) in case of sequence match (or mismatch). It is worth pointing out that the regions corresponding to the structure motifs identified in this manuscript are matched under both SHAPE-only and SHAPE+sequence alignment modes.

3.) Line 80. Writing is ambiguous. The values of 58.15 and 15.47 look to be for FALSE in Fig 2a but come across as for TRUE in the writing.

We thank the reviewer for pointing this mistake out. Indeed, the values refer to the FALSE matches. We have now corrected the main text accordingly.

4.) Where is the ZIKV motif #2 in the figures? Seems like it should be between SI fig #4 and #5.

ZIKV Motif #2 is in Extended Data Figure 2.

5.) End on line 124 might be clearer if worded, “from which SHAPE reagent is employed”

We have now rephrased the sentence as per reviewer’s suggestion.

Reviewer #3 (Remarks to the Author):

This work by Morandi et al., developed a very important tool in identifying the structure homology using SHAPE probing data. Previously, the structure conservation was only measured by sequence alignments. This

SHAPEwarp filled the gap between sequence alignment and experimental probing data. This method will advance the whole research field, allowing the identification of conserved functional structure motif. The methodology development and validations are very clear. The manuscript is well written.

We would like to thank the reviewer for their appreciation of our work.

I only have one comment: The SHAPE profiles for individual RNAs are highly dependent on the sequencing depth. Is there any requirement on the SHAPE reactivity profiles (eg., reads/nt?) for running the SHAPEwarp alignment?

We totally agree with the reviewer. Accurate derivation of reliable SHAPE profiles strongly depends on having sufficient sequencing depth. However, this is a requirement that must be enforced during the SHAPE data analysis steps. As SHAPEwarp works by directly using normalized SHAPE reactivity profiles as input, it is not aware of the original sequencing depth. To ensure the reconstruction of accurate SHAPE profiles and the reliability of the analyses here detailed, we enforced on all the SHAPE-MaP datasets we generated a minimum coverage per-base of 1000X.

REVIEWERS' COMMENTS

Reviewer #1 (Remarks to the Author):

All issues that I have raised were satisfactorily addressed.

Reviewer #2 (Remarks to the Author):

Reviewer 2: I appreciate the efforts the authors took to address my comments. I support publication and have one minor comment.

1.) In response to my Major Comment 1, it would be appreciated if the authors made it clear in the manuscript itself which data is from MaP and which is from stops.

Reviewer #3 (Remarks to the Author):

The authors have fully addressed my comments.

Reviewer #1 (Remarks to the Author):

All issues that I have raised were satisfactorily addressed.

Reviewer #2 (Remarks to the Author):

I appreciate the efforts the authors took to address my comments. I support publication and have one minor comment.

1) In response to my Major Comment 1, it would be appreciated if the authors made it clear in the manuscript itself which data is from MaP and which is from stops.

Reviewer #3 (Remarks to the Author):

The authors have fully addressed my comments.

Authors' response:

We are glad that all reviewers appreciated our efforts to address their comments and to improve the manuscript. Concerning the last request of Reviewer 2, we have now specified in the text, where the icSHAPE dataset is mentioned for the first time, that this is an “RT stop-based readout” (page 4, line 14 of the revised manuscript). For the SHAPE-MaP datasets the fact that they are derived by mutational profiling is self explanatory (because of the “MaP” acronym). We have further indicated that the ZIKV icSHAPE data is RT stop-derived, by adding the following statement at line 14 of page 5 in the revised manuscript, here reported: “For instance, the extensive similarity detected for the ZIKV genomes underscores this point. Indeed, while optimal SHAPEwarp alignment parameters were derived from MaP data, the icSHAPE data for ZIKV genomes was generated using an RT-stop-based readout strategy”.

We would also like to take the occasion to thank the reviewers for their constructive criticisms that helped significantly improving the manuscript.